# MᴇᴍCNN: ᴀ Fʀᴀᴍᴇᴡᴏʀᴋ ғᴏʀ Dᴇᴠᴇʟᴏᴘɪɴɢ Mᴇᴍᴏʀʏ Eғғɪᴄɪᴇɴᴛ Dᴇᴇᴘ Iɴᴠᴇʀᴛɪʙʟᴇ Nᴇᴛᴡᴏʀᴋs

**Sil C. van de Leemput, Jonas Teuwen & Rashindra Manniesing** *
Diagnostic Image Analysis Group
Department of Radiology and Nuclear Medicine
Radboud University Medical Center
6525 GA Nijmegen, The Netherlands

## Aʙsᴛʀᴀᴄᴛ

Reversible operations have recently been successfully applied to classification problems to reduce memory requirements during neural network training. This feature is accomplished by removing the need to store the input activation for computing the gradients at the backward pass and instead reconstruct them on demand. However, current approaches rely on custom implementations of back-propagation, which limits applicability and extendibility. We present MemCNN, a novel PyTorch framework which simplifies the application of reversible functions by removing the need for a customized backpropagation. The framework contains a set of practical generalized tools, which can wrap common operations like convolutions and batch normalization and which take care of the memory management. We validate the presented framework by reproducing state-of-the-art experiments comparing classification accuracy and training time on Cifar-10 and Cifar-100 with the existing state-of-the-art, achieving similar classification accuracy and faster training times.

## 1  Iɴᴛʀᴏᴅᴜᴄᴛɪᴏɴ

Reversible functions, which allow exact retrieval of its input from its output, can reduce memory overhead when used within the context of training neural networks with backpropagation. That is, since only the output requires to be stored, intermediate feature maps can be freed on the forward pass and recomputed from the output on the backward pass when required. Recently, reversible functions have been used with some success to extend the well established residual network (ResNet) for image classification from He et al. (2015) to more memory efficient invertible convolutional neural networks (Gomez et al., 2017; Chang et al., 2017; Jacobsen et al., 2018) showing competing performance on datasets like Cifar-10, Cifar-100 (Krizhevsky & Hinton, 2009) and ImageNet (Deng et al., 2009).

The reversible residual network (RevNet) of Gomez et al. (2017) is a variant on ResNet, which hooks into its sequential structure of residual blocks and replaces them with reversible blocks, that creates an explicit inverse for the residual blocks based on the equations from Dinh et al. (2014) on nonlinear independent components estimation. The reversible block takes arbitrary non-linear functions $\mathcal{F}$ and $\mathcal{G}$ and renders them invertible. Their experiments show that RevNet scores similar classification performance on Cifar-10, Cifar-100, and ImageNet, with less memory overhead.

Reversible architectures like RevNet have subsequently been studied in the framework of ordinary differential equations (ODE) (Chang et al., 2017). Three reversible neural network based on Hamiltonian systems are proposed, which are similar to the RevNet, but have a specific choice for the non-linear functions $\mathcal{F}$ and $\mathcal{G}$ which are shown stable during training within the ODE framework on Cifar-10 and Cifar-100.

The i-RevNet architecture extends the RevNet architecture by also making the downscale operations invertible Jacobsen et al. (2018), effectively creating a fully invertible architecture up until the last layer, while still showing good classification accuracy compared to ResNet on ImageNet. One

---

*Correspondence to `sil.vandeleemput@radboudumc.nl`

particularly interesting finding shows that bottlenecks are not a necessary condition for training neural networks, which shows that the study of invertible networks can lead to a better understanding of neural networks training in general.

The different reversible architectures proposed in the literature (Gomez et al., 2017; Chang et al., 2017; Jacobsen et al., 2018) have all been modifications of the ResNet architecture and all have been implemented in TensorFlow (Abadi et al., 2015). However, these implementations rely on custom backpropagation, which limits creating novel invertible networks and application of the concepts beyond the application architecture. Our proposed framework MemCNN overcomes this issue by being compatible with the default backpropagation facilities of PyTorch. Furthermore, PyTorch offers convenient features over other deep learning frameworks like a dynamic computation graph and simple inspection of gradients during backpropagation, which facilitates inspection of invertible operations in neural networks.

In this work we present MemCNN [1] a novel PyTorch (Paszke et al., 2017) implementation which simplifies the use of reversible functions by removing the need for a customized backpropagation. MemCNN provides tools to drop-in memory saving reversible functions within conventional PyTorch neural networks. Furthermore, it provides wrappers to convert arbitrary non-linear functions to memory saving reversible functions. We have validated the presented framework by implementing two state-of-the-art architectures (ResNet and RevNet) utilizing MemCNN, which are included in the GitHub repository, and compare them to existing state-of-the-art implementations in TensorFlow (Abadi et al., 2015) on the Cifar-10 and Cifar-100 classification tasks on accuracy and training time. Our framework was found to achieve similar classification accuracy and faster training times. Validation experiments described in this work are included in the framework as well.

## 2 METHODOLOGY

### 2.1 THE REVERSIBLE BLOCK

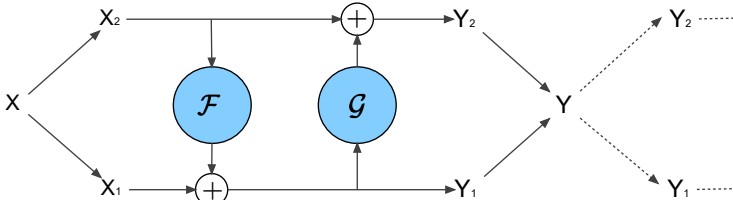

Figure 1: Graphical representation of the reversible block. First, input $x$ is partitioned in two same shaped sets $x_1$ and $x_2$ after which $\mathcal{F}$ and $\mathcal{G}$ are applied to obtain $y_1$ and $y_2$, which are finally concatenated to form output $y$. The reversible block can be chained by subsequent reversible blocks or be combined with regular functions operating on an unpartitioned set, e.g. $f(x) = y$.

The core reversible operation of MemCNN follows the equations of Gomez et al. (2017); Dinh et al. (2014), since they support a template for specific reversible implementations through $\mathcal{F}$ and $\mathcal{G}$. However, for incorporation of the equations with the default automatic differential system of PyTorch we have encapsulated the equations as a reversible block function $R_{\mathcal{F},\mathcal{G}}$ which partitions its input $x$ into two sets $x_1$ and $x_2$ of equal shape and computes the concatenation $y = (y_1, y_2)$ using equation 1. Figure 1 shows a graphical representation of the reversible block function $R_{\mathcal{F},\mathcal{G}}$.

$$R_{\mathcal{F},\mathcal{G}}(x) := (y_1, y_2) \text{ with } x = (x_1, x_2) \qquad R_{\mathcal{F},\mathcal{G}}^{-1}(y) := (x_1, x_2) \text{ with } y = (y_1, y_2)$$
$$y_1 = x_1 + \mathcal{F}(x_2) \qquad (1) \qquad x_1 = y_1 - \mathcal{F}(x_2) \qquad (2)$$
$$y_2 = x_2 + \mathcal{G}(y_1) \qquad x_2 = y_2 - \mathcal{G}(y_1)$$

Where $\mathcal{F}$ and $\mathcal{G}$ can be arbitrary nonlinear functions like convolutions, ReLUs, etc., as long as they have matching input and output shapes, i.e. $shape(x_1) = shape(x_2) = shape(y_1) = shape(y_2)$. Using the same equations, $x$ can be approximated from $y$ by $x = R_{\mathcal{F},\mathcal{G}}^{-1}(y)$ rewriting the formulas of equation 1 to its inverse in equation 2.

---

[1] MemCNN is available at: https://github.com/silvandeleemput/memcnn

Table 1: Accuracy (acc.) and training time (time, in hours:minutes) comparison of the PyTorch implementation (MemCNN) versus the Tensorflow implementation from Gomez et al. (2017) on Cifar-10 and Cifar-100. (Krizhevsky & Hinton, 2009)

| | Tensorflow | | | | PyTorch | | | |
| | Cifar-10 | | Cifar-100 | | Cifar-10 | | Cifar-100 | |
| Model | acc. | time | acc. | time | acc. | time | acc. | time |
|---|---|---|---|---|---|---|---|---|
| resnet-32 | 92.74 | 2:04 | 69.10 | 1:58 | 92.86 | 1:51 | 69.81 | 1:51 |
| resnet-110 | 93.99 | 4:11 | 73.30 | 6:44 | 93.55 | 2:51 | 72.40 | 2:39 |
| resnet-164 | 94.57 | 11:05 | 76.79 | 10:59 | 94.80 | 4:59 | 76.47 | 3:45 |
| revnet-38 | 93.14 | 2:17 | 71.17 | 2:20 | 92.54 | 1:10 | 69.33 | 1:40 |
| revnet-110 | 94.02 | 6:59 | 74.00 | 7:03 | 93.25 | 3:43 | 72.24 | 3:44 |
| revnet-164 | 94.56 | 13:09 | 76.39 | 13:12 | 93.40 | 7:19 | 74.63 | 7:21 |

## 2.2 IMPLEMENTATION DETAILS

The entire reversible block has been implemented as a 'torch.nn.Module' which can wrap Pytorch modules for arbitrary functions $\mathcal{F}$ and $\mathcal{G}$ and has a default forward and backward pass implemented using a 'torch.autograd.Function' for support with the automatic differentiation system of PyTorch.

The forward pass follows equation 1 to compute $R_{\mathcal{F},\mathcal{G}}(x) = y$ and afterwards frees $x$ from memory by default. The backward pass first checks if $x$ is stored in memory, if not, $x$ is computed through equation 2, Subsequently, $x'$ derivatives for $x$, $\mathcal{F}'_W$ derivatives for the weights of $\mathcal{F}$, and $\mathcal{G}'_W$ derivatives for the weights of $\mathcal{G}$ are computed by using the PyTorch 'autograd' solver.

## 3 EXPERIMENTS AND RESULTS

To validate MemCNN, we reproduced the experiments from Gomez et al. (2017) on Cifar-10 and Cifar-100 (Krizhevsky & Hinton, 2009) using their Tensorflow (Abadi et al., 2015) implementation on GitHub [2], and made a direct comparison with our PyTorch implementation on accuracy and train time. We have tried to keep all the experimental settings, like data loading, loss function, train procedure, and training parameters, as similar as possible. All experiments were performed on a single NVIDIA GeForce GTX 1080 with 8GB of RAM. The results are listed in Table 1. Model performance of our PyTorch implementation obtained similar accuracy to the TensorFlow implementation with less training time on Cifar-10 and Cifar-100.

## 4 DISCUSSION AND CONCLUSION

We have presented MemCNN, a novel PyTorch framework, for applying reversible operations for neural networks. It shows similar accuracy on Cifar-10 and Cifar-100 datasets with the current state-of-the-art method for reversible operations in Tensorflow and provides overall faster training times. The main features of the framework are: smooth integration of reversible functions with other non-reversible functions by removing of the need for a custom back propagation and simple wrapping of non-invertible non-linear functions, and easy gradient inspection through Pytorch facilities.

Although these results validate our framework we would like to shed some light on potential future applications and improvements. We would like to provide other reversible architectures like i-RevNet (Jacobsen et al., 2018) and the Hamiltonian-network (Chang et al., 2017) for comparison. Furthermore, we would like to work around the limitations of shape equality for the input and output for the reversible block function and its functions $\mathcal{F}$ and $\mathcal{G}$. To our opinion, the presented framework, which can be found on GitHub, facilitates the study of invertible functions in the context of neural networks.

---

[2]https://github.com/renmengye/revnet-public

ACKNOWLEDGMENTS

This work was supported by research grants from the Netherlands Organization for Scientific Research (NWO), the Netherlands and Canon Medical Systems Corporation, Japan.

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
