# OpenReview forum: "MemCNN: a Framework for Developing Memory Efficient Deep Invertible Networks"
_ICLR.cc/2018/Workshop — Accept_

### Official Review · AnonReviewer1 · 2018-03-01
**Faster Implementation of RevNets**

**Rating:** 6
**Confidence:** 3

**Review:**

This paper describes a software library based on PyTorch that provides easier mechanism to implement reversible neural network architectures. the main contribution seem to be a general API that does some clever memory handling. By utilizing the PyTorch's builtin autograd mechanism, it is shown to be faster than Tensorflow based implementation that uses custom gradient function.

It is of potential interest to the community to use this tool to build their own reversible architectures and do related research. However, this paper could potentially be improved

- As a paper describing a library, it might worse spending more time describing the main contributions and novelty in the design and implementation of this software (than describing the literature of RevNet researches).

- It might be good to have brief examples (maybe code) showing how end users use this.

- MemCNN is a confusing name. It sounds like something related to differentiable memory architectures.

---

### Official Review · AnonReviewer3 · 2018-03-10
**Less information about the framework**

**Rating:** 4
**Confidence:** 5

**Review:**

The paper describes a new framework on PyTorch to implement arbitrary reversible connections. It may become a useful functionality to implement those architectures. However, the paper describes only a bit information not enough to argue the advantage of the proposed framework.
Table 1 has less information about differences in time consumption between TensorFlow and PyTorch implementations, because it also looks like the effect due to only the difference of the base framework. At least authors should add experiments about all models with "naive implementations" on PyTorch.
There is no information about the usage of the framework on both the paper and the repository, which is essentially important for this kind of papers. E.g., small (a few lines) sample codes, and descriptions of how the code works can greatly help potential users of the framework.

Other comments:

* The framework name "MemCNN" does not express the actual functionality: implementing reversible connections, and reconsidering the name is recommended for the community.
* The order of equations about $x_1$ and $x_2$ in Eq. (2) should be reversed to maintain the actual order of the calculation (like what written in [Gomez et al., '17]).
* Table 1 looks a bit confusing because results about datasets and acc/time appears alternately. Separating the table according to at least an axis (dataset or acc/time) can help readers to comprehend the results.

---

### Official Review · AnonReviewer2 · 2018-03-12
**Nice python framework implementation of reversible functions in CNN. Useful for DL practicioners.**

**Rating:** 7
**Confidence:** 3

**Review:**

Before providing my review I need to add that this is not my area of expertise. As such I will mainly review this work regarding its quality, clarity and significance (the latter in a way explained in what follows).

Reversible operations as part of a NN are shown to reduce memory requirements during training. As such, their application as part of the NN training design has attracted lots of attention. Current approaches implement the reversible functions using backpropagation which prohibits the application of novel invertible networks. The authors present the MemCNN, a PyTorch implementation which implements reversibility without the need for customized backpropagation. Their implementation is based on the equations of Gomez et all and Dinh et al. They successfully managed to incorporate the ideas into an efficient PyTorch framework. They support their implementation by providing results on two distinct classification tasks where the framework is compared to the state-of-the-art Tensor-Flow implementation. Both implementations seem to be comparable in accuracy. The MemCNN outperforms in training time though.

I am pleased with the quality and the clarity of the work presented here. The authors did a good job presenting their contribution in a clear way by stating the motivation and plugging their implementation in the picture. I think their contribution is significant for the DL practitioners; a Python framework that allows for less training time is attractive.

---

### Decision · Program_Chairs · 2018-03-20
**ICLR 2018 Workshop Acceptance Decision**

**Decision:**

Accept

**Comment:**

Congratulations, your paper was accepted to the ICLR workshop.